# Data-Efficient Learning with Neural Programs

**Alaia Solko-Breslin, Seewon Choi, Ziyang Li, Neelay Velingker,**
**Rajeev Alur, Mayur Naik, Eric Wong**
University of Pennsylvania
{alaia,seewon,liby99,neelay,alur,mhnaik,exwong}@seas.upenn.edu

## Abstract

Many computational tasks can be naturally expressed as a composition of a DNN followed by a program written in a traditional programming language or an API call to an LLM. We call such composites "neural programs" and focus on the problem of learning the DNN parameters when the training data consist of end-to-end input-output labels for the composite. When the program is written in a differentiable logic programming language, techniques from neurosymbolic learning are applicable, but in general, the learning for neural programs requires estimating the gradients of black-box components. We present an algorithm for learning neural programs, called ISED, that only relies on input-output samples of black-box components. For evaluation, we introduce new benchmarks that involve calls to modern LLMs such as GPT-4 and also consider benchmarks from the neurosymbolic learning literature. Our evaluation shows that for the latter benchmarks, ISED has comparable performance to state-of-the-art neurosymbolic frameworks. For the former, we use adaptations of prior work on gradient approximations of black-box components as a baseline, and show that ISED achieves comparable accuracy but in a more data- and sample-efficient manner. [1]

## 1 Introduction

Many computational tasks cannot be solved by neural perception alone but can be naturally expressed as a composition of a neural model $M_\theta$ followed by a program $P$ written in a traditional programming language or an API call to a large language model (LLM). We call such composites "neural programs" and study the problem of learning neural programs in an end-to-end manner with a focus on data and sample efficiency. One problem that is naturally expressed as a neural program is scene recognition [29], where $M_\theta$ classifies objects in an image and $P$ prompts GPT-4 to identify the room type given these objects (Fig. 1).

Neurosymbolic learning [2] is one instance of neural program learning in which $P$ takes the form of a logic program. DeepProbLog (DPL) [14] and Scallop [13] are frameworks that extend ProbLog and Datalog, respectively, to ensure that the symbolic component $P$ is differentiable. This differentiability requirement is what facilitates learning in many neurosymbolic learning frameworks. There are also abductive learning frameworks that do not explicitly differentiate programs. Instead, they require that the symbolic component expose a method for abducing the function's inputs for a given output, often using Prolog for the symbolic component as a result [6, 23]. While logic programming languages are expressive enough for these frameworks to solve tasks such as sorting [14], visual question answering [13], and path planning [23], they offer restricted features and a narrow range of libraries, making them incompatible with calls to arbitrary APIs or to modern LLMs.

Learning neural programs when $P$ is not expressed as a logic program is a difficult problem because gradients across black-box programs cannot be computed explicitly. One possible solution is to use REINFORCE [26] to sample symbols from distributions predicted by $M_\theta$ and compute the expected

---

[1]Code is available at https://github.com/alaiasolkobreslin/ISED

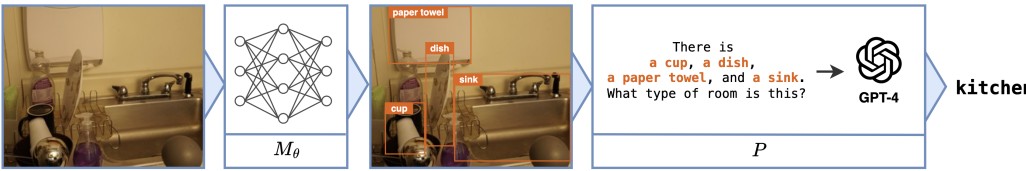

Figure 1: Neural program decomposition for scene recognition.

reward using the output label. However, REINFORCE is not sample-efficient as it produces a weak learning signal, especially when applied to programs with a large number of inputs. There are other REINFORCE-based methods that can be applied to the neural program learning setting, namely IndeCateR [21] and Neural Attention for Symbolic Reasoning (NASR) [5]. However, IndeCateR struggles with sample efficiency despite providing lower variance than REINFORCE, and NASR performs poorly when intermediate labels are unavailable for pretraining. Another possible solution is Approximate Neurosymbolic Inference (A-NeSI) [24], which trains a neural network to estimate the gradient of $P$, but learning the surrogate neural network becomes more difficult as the complexity of $P$ increases. Moreover, the additional neural models in the learning framework in A-NeSI results in data inefficiency.

In this paper, we propose an algorithm for learning neural programs, based on reinforcement learning, which is compatible with arbitrary programs. Our approach, called ISED (Infer-Sample-Estimate-Descend), yields a framework that expands the applicability of neural program learning frameworks by providing a data- and sample-efficient method of training neural models with randomly initialized weights. ISED uses outputs of $M_\theta$ as a probability distribution over inputs of $P$ and samples representative symbols $u$ from this distribution. ISED then computes outputs $v$ of $P$ corresponding to these symbols. The resulting symbol-output pairs can be viewed as a symbolic program consisting of clauses of the form `if symbol = u then output = v` summarizing $P$. The final step is to estimate the gradient across this symbolic summary, inspired by ideas from the neurosymbolic learning literature, to propagate loss across the composite model.

Our evaluation considers 16 neural program benchmark tasks. Our results show that ISED outperforms purely neural networks and CLIP [19] on neural program tasks involving GPT-4 calls. Additionally, ISED outperforms neurosymbolic methods on 9 of the 14 benchmarks tasks that can be encoded in logic programming languages. ISED is also the top performer on 8 out of the 16 benchmark tasks when compared to REINFORCE-based and black-box gradient estimation baselines. Furthermore, we show that ISED is more data- and sample-efficient than baseline methods.

In summary, the main contributions of this paper are as follows: 1) we introduce neural programs as a generalization of neurosymbolic programs, 2) we introduce new tasks involving neural programs that use Python and calls to GPT-4 called neuroPython and neuroGPT programs, respectively, 3) we present ISED, a general algorithm for data- and sample-efficient learning with neural programs, and 4) we conduct a thorough evaluation using existing techniques against a diverse set of benchmarks.

## 2 Neural Programs

**Problem Statement.** In the neural program learning setting, we attempt to optimize model parameters $M_\theta$ which are being supervised by a fixed program $P$. Specifically, we are given a training dataset $\mathcal{D}$ of length $N$ containing input-output pairs, i.e., $\mathcal{D} = \{(x_1, y_1), \dots (x_N, y_N)\}$. Each $x_i$ represents unstructured data (e.g., image data) whose corresponding structured data (intermediate labels) are not given. Each $y_i$ is the result of applying $P$ to the structured data corresponding to $x_i$. Given a loss function $\mathcal{L}$, we want to minimize the loss of $\mathcal{L}(P(M_\theta(x_i)), y_i)$ for each $(x_i, y_i)$ pair in order to optimize $\theta$. Loss minimization is straightforward when there is some mechanism for automatically differentiating programs, but we focus on the setting of optimizing $\theta$ without assuming the differentiability of $P$. We now introduce three motivating applications that can be framed in this

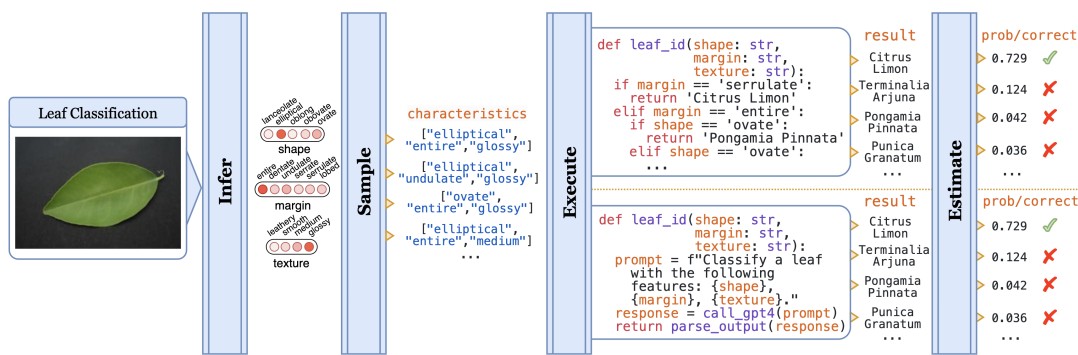

Figure 2: Illustration of our inference pipeline for the leaf classification task. `leaf_id` can be written with a decision tree (top program) or with a call to GPT-4 (bottom program).

setting, namely classifying images of leaves [9], scene recognition [29], and hand-written formula evaluation (HWF) [12].

**Leaf Classification.** We consider a real-world example that deals with the problem of classifying leaf images. Traditional neural methods predict the species directly, without explicit notion of leaf features such as margin, shape, and texture, resulting in solutions that are data-inefficient, inaccurate, and harder to understand.

We instead present a neural programming solution, making use of leaf classification decision trees [22]. These decision trees allow identifying plant species based on the visible characteristics of their leaves. Here, the neural model takes a leaf image and predicts its shape, margin, and texture. The program can then be written in two ways: one implementation involves encoding the decision tree in Python; another involves constructing a prompt using the predicted leaf features and calling GPT-4 (see Fig. 2). The latter is possible because ISED allows the use of black-box programs, so programs can also use state-of-the-art foundation models such as GPT-4 for computation.

**Scene Recognition.** The goal of this task is to classify images according to their room types. The model receives an image from a scene dataset [16] and predicts among the 9 different room types: bedroom, bathroom, dining room, living room, kitchen, lab, office, house lobby, and basement.

The traditional neural solution trains a convolutional neural network that directly predicts the room type. On the other hand, the neural program solution decomposes the task into detecting objects in the scene and identifying the room type based on those objects. We use an off-the-shelf object detection model YOLOv8 [20] and finetune it with a custom convolutional neural network to output labels related to scene recognition. We then make a GPT-4 call to predict the most likely room type given the list of detected objects.

**Hand-written Formula.** In this task, a model is given a list of hand-written symbols containing digits (0-9) and operators ($+$, $-$, $\times$, and $\div$) [12]. The dataset contains length 1-7 formulas free of syntax or divide-by-zero errors. The model is trained with supervision on the evaluated floating-point result without the label of each symbol. Since inputs are combinatorial and results are rational numbers, end-to-end neural methods struggle with accuracy. Meanwhile, neurosymbolic methods for this task either use specialized algorithms [12] or handcrafted differentiable programs [13].

With ISED, the program can be written in just a few lines of Python. It takes in a list of characters representing symbols, and simply invokes the Python `eval` function on the joined expression string. The `hwf` evaluation function can be used just like any other PyTorch [17] module since ISED internally performs sampling and probability estimation to estimate the gradient.

## 3   Learning Neural Programs

In this section, we present the intuition behind ISED and the values it approximates. Next, we introduce the programming interface for ISED, which lays the groundwork for presenting the algorithm. We then formally describe the steps of ISED.

## 3.1 ISED Overview

Assuming $P$ is a black-box, we can collect symbol-output samples $(u, v)$ from $P$. Such collection of samples can be viewed as a *summary* logic program consisting of rules of the form `if` $r = u$ `then` $y = v$. For instance, in the task of adding two digits $r_1$ and $r_2$, one rule of the logic program would be $r_1 = 1 \wedge r_2 = 2 \rightarrow y = 3$. Techniques from neurosymbolic literature via exact or approximate weighted model counting (WMC) [10] can then be used for computing the gradient across such a summary of $P$. However, having the complete summary of all combinations of symbols is not feasible for a black-box $P$. ISED samples symbols from the probability distribution predicted by the neural network $M_\theta$, evaluates $P$ on each sample, and takes the gradient across this partial summary of $P$. This is a good approximation of the complete summary since it is likely to contain symbols with high probability, which contribute the most in exact computation.

This approach differs from REINFORCE in how it differentiates through this summary of $P$. RE-INFORCE rewards sampled symbols that resulted in the correct output through optimizing the log probability of each symbol, weighted by reward values. This weighted-sum style estimation provides a weaker learning signal compared to WMC used by ISED, making learning harder for REINFORCE as the number of inputs to $P$ increases. See Appendix A for further details.

## 3.2 Preliminaries and Programming Interface

ISED allows programmers to write black-box programs that operate on diverse structured inputs and outputs. To allow such programs to interact with neural networks, we define an interface named *structural mapping*. This interface serves to 1) define the data-types of black-box programs' input and output, 2) marshall and un-marshall data between neural networks and logical black-box functions, and 3) define the loss. We define a *structural mapping* $\tau$ as either a discrete mapping (with $\Sigma$ being the set of all possible elements), a floating point, a permutation mapping with $n$ possible elements, a tuple of mappings, or a list of up to $n$ elements. We define $\tau$ inductively as follows:

$$\tau ::= \text{DISCRETE}(\Sigma) \mid \text{FLOAT} \mid \text{PERMUTATION}_n \mid \text{TUPLE}(\tau_1, \ldots, \tau_m) \mid \text{LIST}_n(\tau)$$

Using this, we may further define data-types such as $\text{INTEGER}_j^k = \text{DISCRETE}(\{j, \ldots, k\})$, $\text{DIGIT} = \text{INTEGER}_0^9$, and $\text{BOOL} = \text{DISCRETE}(\{\text{true}, \text{false}\})$. These types give ISED the flexibility learn neural programs with diverse types of inputs and outputs, e.g., $\text{PERMUTATION}_n$ input and output types for integer list sorting and $\text{LIST}_9(\text{LIST}_9(\text{DIGIT}))$ for sudoku solving.

We also define a *black-box program $P$* as a function $(\tau_1, \ldots, \tau_m) \rightarrow \tau_o$, where $\tau_1, \ldots, \tau_m$ are the input types and $\tau_o$ is the output type. For example, the structural input mapping for the hand-written formula task is $\text{LIST}_7(\text{DISCRETE}(\{0, \ldots, 9, +, -, \times, \div\}))$, and the structural output mapping is $\text{FLOAT}$. The mappings suggest that the program takes a list of length up to 7 as input, where each element is a digit or an arithmetic operator, and returns a floating point number.

There are two interpretations of a structural mapping: the set interpretation $\text{SET}(\tau)$ represents a mapping with defined values, e.g., a digit with value 8; the tensor interpretation $\text{DIST}(\tau)$ represents a mapping where each value is associated with a probability distribution, e.g., a digit that is 1 with probability 0.6 and 7 with probability 0.4. We use the set interpretation to represent structured program inputs that can be passed to a black-box program and the tensor interpretation to represent probability distributions for unstructured data and program outputs. These two interpretations are defined for the different structural mappings in Table 1.

Table 1: Set and tensor interpretations of different structural mappings.

| Mapping ($\tau$) | Set Interpretation ($\text{SET}(\tau)$) | Tensor Interpretation ($\text{DIST}(\tau)$) |
|---|---|---|
| $\text{DISCRETE}_\Sigma$ | $\Sigma$ | $\{\vec{v} \mid \vec{v} \in \mathbb{R}^{|\Sigma|}, v_i \in [0, 1], i \in 1 \ldots |\Sigma|\}$ |
| $\text{FLOAT}$ | $\mathbb{R}$ | n/a |
| $\text{PERMUTATION}_n$ | $\{\rho \mid \rho \text{ is a permutation of } [1, \ldots, n]\}$ | $\{[\vec{v_1}, \ldots, \vec{v_n}] \mid \vec{v_i} \in \mathbb{R}^n, v_{i,j} \in [0, 1], i \in 1 \ldots n\}$ |
| $\text{TUPLE}(\tau_1, \ldots, \tau_m)$ | $\{(a_1, \ldots, a_m) \mid a_i \in \text{SET}(\tau_i)\}$ | $\{(a_1, \ldots, a_m) \mid a_i \in \text{DIST}(\tau_i)\}$ |
| $\text{LIST}_n(\tau')$ | $\{[a_1, \ldots, a_j] \mid j \leq n, a_i \in \text{SET}(\tau')\}$ | $\{[a_1, \ldots, a_j] \mid j \leq n, a_i \in \text{DIST}(\tau')\}$ |

In order to represent the ground truth output as a distribution to be used in the loss computation, there needs to be a mechanism for transforming $\text{SET}(\tau)$ mappings into $\text{DIST}(\tau)$ mappings. For

this purpose, we define a *vectorize* function $\delta_\tau : (\text{SET}(\tau), 2^\tau) \to \text{DIST}(\tau)$ for the different output mappings $\tau$ in Table 2. When considering a datapoint $(x, y)$ during training, ISED samples many symbols and obtains a list of outputs $\hat{y}$. The vectorizer then takes the ground truth $y$ and the outputs $\hat{y}$ as input and returns the equivalent distribution interpretation of $y$. While $\hat{y}$ is not used by $\delta_\tau$ in most cases, we include it as an argument so that FLOAT output mappings can be discretized, which is necessary for vectorization. For example, if the inputs to the vectorizer for the hand-written formula task are $y = 2.0$ and $\hat{y} = [1.0, 3.5, 2.0, 8.0]$, then it would return $[0, 0, 1, 0]$.

Table 2: Vectorize and aggregate functions of different structural mappings.

| Mapping ($\tau$) | Vectorizer ($\delta_\tau(y, \hat{y})$) | Aggregator ($\sigma_\tau(\hat{r}, \hat{p})$) |
|---|---|---|
| DISCRETE$_n$ | $e^{(y)}$ with dim $n$ | $\hat{p}[\hat{r}]$ |
| FLOAT | $[\mathbf{1}_{y=\hat{y}_i} \text{ for } i \in [1, \ldots, \text{length}(\hat{y})]]$ | n/a |
| PERMUTATION$_n$ | $[\delta_{\text{DISCRETE}_n}(y[i]) \text{ for } i \in [1, \ldots, n]]$ | $\otimes_{i=1}^n \sigma_{\text{DISCRETE}_n}(\hat{r}[i], \hat{p}[i])$ |
| TUPLE$(\tau_1, \ldots, \tau_m)$ | $[\delta_{\tau_i}(y[i]) \text{ for } i \in [1, \ldots, m]]$ | $\otimes_{i=1}^m \sigma_{\tau_i}(\hat{r}[i], \hat{p}[i])$ |
| LIST$_n(\tau')$ | $[\delta_{\tau'}(a_i) \text{ for } a_i \in y]$ | $\otimes_{i=1}^n \sigma_{\tau'}(\hat{r}[i], \hat{p}[i])$ |

We also require a mechanism to aggregate the probabilities of sampled symbols that resulted in a particular output. With this aim, we define an *aggregate* function $\sigma_\tau : (\text{SET}(\tau), \text{DIST}(\tau)) \to \mathbb{R}$ for different input mappings $\tau$ in Table 2. ISED aggregates probabilities either by taking their minimum or their product, and we denote both operations by $\otimes$. The aggregator takes as input sampled symbols $\hat{r}$ and neural predictions $\hat{p}$ from which $\hat{r}$ was sampled. It gathers values in $\hat{p}$ at each index in $\hat{r}$ and returns the result of $\otimes$ applied to these values. For example, suppose we use min as the aggregator $\otimes$ for the hand-written formula task. Then if $\otimes$ takes $\hat{r} = [1, +, 1]$ and $\hat{p}$ as inputs where $\hat{p}[0][1] = 0.1$, $\hat{p}[1][+] = 0.05$, and $\hat{p}[2][1] = 0.1$, it would return $0.05$.

## 3.3 Algorithm

We now formally present the ISED algorithm. For a given task, there is a black-box program $P$, taking $m$ inputs, that operates on structured data. Let $\tau_1, ..., \tau_m$ be the mappings for these inputs and $\tau_o$ the mapping for the program's output. We write $P$ as a function from its input mappings to its output mapping: $P : (\tau_1, ..., \tau_m) \to \tau_o$. For each unstructured input $i$ to the program, there is a neural model $M_{\theta_i}^i : x_i \to \text{DIST}(\tau_i)$. $S$ is a sampling strategy (e.g., categorical sampling) that samples symbols using the outputs of a neural model, and $k$ is the number of samples to take for each training example. There is also a loss function $\mathcal{L}$ whose first and second arguments are the predicted and ground truth values respectively. We present the pseudocode of the algorithm in Algorithm 1 and describe its steps with the hand-written formula task:

**Infer.** The training pipeline starts with an example from the dataset, $(x, y) = ([\diagup, +, 2], 3.0)$, and uses a CNN to predict these images, as shown on lines 3-4. ISED initializes $\hat{p} = M_\theta(x)$.

**Sample.** ISED samples $\hat{r}$ from $\hat{p}$ for $k$ iterations using sampling strategy $S$. For each sample $j$, the algorithm initializes $\hat{r}_j$ to be the sampled symbols, as shown on lines 6-9. To continue our example, suppose ISED initializes $\hat{r}_j = [7, +, 2]$ for sample $j$. The next step is to execute the program on $\hat{r}_j$, as shown on line 10, which in this example means setting $\hat{y}_j = P(\hat{r}_j) = 9.0$.

**Estimate.** In order to compute the prediction value to use in the loss function, ISED must consider each output $y_l$ in the output mapping and accumulate the aggregated probabilities for all sampled symbols that resulted in the output $y_l$. We specify $\otimes$ as the min function, and $\oplus$ as the max function in this example. Note that ISED requires that $\otimes$ and $\oplus$ represent either min and max or mult and add respectively. We refer to these two options as the min-max and add-mult semirings. We define an *accumulate* function $\omega$ that takes as input an element of the output mapping $y_l$, sampled outputs $\hat{y}$, sampled symbols $\hat{r}$, and predicted input distributions $\hat{p}$. The accumulator performs the $\oplus$ operation on aggregated probabilities for elements of $\hat{y}$ that are equal to $y_l$ and is defined as follows:

$$\omega(y_l, \hat{y}, \hat{r}, \hat{p}) = \oplus_{j=1}^k \mathbf{1}_{\hat{y}_j = y_l} \sigma_{\tau_o}(\hat{r}_j, \hat{p}_j)$$

Continuing our example, suppose, among the samples, there are two symbolic combinations ($[7, +, 2]$ and $[3, *, 3]$) that resulted in the output $9.0$. Let us say that these sets of symbols had probabilities $[0.3, 0.8, 0.8]$ and $[0.1, 0.1, 0.1]$, respectively. Then the result of the probability aggregation for $y_l = 9.0$ would be $\omega(9.0, \hat{y}, \hat{r}, \hat{p}) = \texttt{max}(\texttt{min}([0.3, 0.8, 0.8]), \texttt{min}([0.1, 0.1, 0.1])) = 0.3$.

---

**Algorithm 1** ISED training pipeline

---

**Require:** $P$ is the black-box program $(\tau_1, \ldots, \tau_m) \to \tau_o$, $M_{\theta_i}^i$ the neural model $x_i \to \text{DIST}(\tau_i)$ for each $\tau_i$, $S$ the sampling strategy, $k$ the sample count, $\mathcal{L}$ the loss function, and $\mathcal{D}$ the dataset.

1: **procedure** TRAIN
2:  **for** $((x_1, \ldots x_m), y) \in \mathcal{D}$ **do**
3:    **for** $i \in 1 \ldots m$ **do**
4:      $\hat{p}[i] \leftarrow M_{\theta_i}^i(x_i)$                                                    ▷ **Infer**
5:    **end for**
6:    **for** $j \in 1 \ldots k$ **do**
7:      **for** $i \in 1 \ldots m$ **do**
8:        Sample $\hat{r}_j[i]$ from $\hat{p}[i]$ using $S$                          ▷ **Sample**
9:      **end for**
10:      $\hat{y}_j \leftarrow P(\hat{r}_j)$
11:    **end for**
12:    $\hat{w} \leftarrow \text{normalize}([\omega(y_l, \hat{y}, \hat{r}, \hat{p}) \text{ for } y_l \in \tau_o \text{ (or } y_l \in \hat{y})])$        ▷ **Estimate**
13:    $w \leftarrow \delta(y, \hat{y})$
14:    $l \leftarrow \mathcal{L}(\hat{w}, w)$
15:    Compute $\frac{\partial l}{\partial \theta}$ by performing back-propagation on $l$
16:    Optimize $\theta$ based on $\frac{\partial l}{\partial \theta}$                                        ▷ **Descend**
17:  **end for**
18: **end procedure**

---

ISED then sets $\tilde{w} = [\omega(y_l, \hat{y}, \hat{r}, \hat{p}) \text{ for } y_l \in \tau_o]$ in the case where $\tau_o$ is not FLOAT. When $\tau_o$ is FLOAT, as for hand-written formula, it only considers $y_l \in \hat{y}$. Next, it performs $L_2$ normalization over each element in $\tilde{w}$ and sets $\hat{w}$ to this result. To initialize the ground truth vector, it sets $w = \delta(y, \hat{y})$. ISED then initializes $l = \mathcal{L}(\hat{w}, w)$ and computes $\frac{\partial l}{\partial \theta_i}$ for each input $i$. These steps are shown on lines 12-15. In our running example, since $9.0$ is an incorrect output, the probability of the first symbol being equal to 7 (instead of the correct answer 1) will be penalized while the probabilities for predicting other symbols are unchanged.

**Descend.** The last step is shown on line 16, where the algorithm optimizes $\theta_i$ for each input $i$ based on $\frac{\partial l}{\partial \theta_i}$ using a stochastic optimizer (e.g., Adam optimizer). This completes the training pipeline for one example, and the algorithm returns all final $\theta_i$ after iterating through the entire dataset.

## 4 Evaluation

In this section, we evaluate ISED and aim to answer the following research questions:

**RQ1:** How does ISED compare to state-of-the-art neurosymbolic, REINFORCE-based, and gradient estimation baselines in terms of accuracy?
**RQ2:** What is the sample efficiency of ISED when compared to REINFORCE-based algorithms?
**RQ3:** How data-efficient is ISED compared to neural gradient estimation methods?

### 4.1 Benchmark Tasks: NeuroGPT, NeuroPython, and Neurosymbolic

We first introduce two new neural program learning benchmarks which both contain a program component that can make a call to GPT-4. We call such models neuroGPT programs.

**Leaf Classification.** In this task, we use a dataset, which we call LEAF-ID, containing leaf images of 11 different plant species [4], containing 330 training samples and 110 testing samples. We define custom DISCRETE types MARGIN, SHAPE, TEXTURE. With this, we define LEAF-TRAITS = TUPLE(MARGIN, SHAPE, TEXTURE) and LEAF-OUTPUT to be the DISCRETE set of 11 plant species in the dataset. Neural program solutions either prompt GPT-4 (GPT leaf) or use a decision tree (DT leaf).

**Scene Recognition.** We use a dataset containing scene images from 9 different room types [16], consisting of 830 training examples and 92 testing examples. We define custom types OBJECTS and SCENES to be DISCRETE set of 45 objects and 9 room types, respectively. We freeze the parameters

Table 3: Performance on selected benchmarks. "TO" means time-out, and "N/A" means the task could not be programmed in the framework. Methods are divided (from top to bottom) by neurosymbolic, black-box gradient estimation, and REINFORCE-based.

| | Accuracy (%) | | | | | | | |
|---|---|---|---|---|---|---|---|---|
| **Method** | $sum_2$ | $sum_3$ | $sum_4$ | HWF | DT leaf | GPT leaf | scene | sudoku |
| DPL | 95.14 | 93.80 | TO | TO | 39.70 | N/A | N/A | TO |
| Scallop | 91.18 | 91.86 | 80.10 | 96.65 | 81.13 | N/A | N/A | TO |
| A-NeSI | **96.66** | 94.39 | 78.10 | 3.13 | 78.82 | 72.40 | 61.46 | 26.36 |
| REINFORCE | 74.46 | 19.40 | 13.84 | 88.27 | 40.24 | 53.84 | 12.17 | 79.08 |
| IndeCateR | 96.48 | 93.76 | 92.58 | 95.08 | 78.71 | 69.16 | 12.72 | 66.50 |
| NASR | 6.08 | 5.48 | 4.86 | 1.85 | 16.41 | 17.32 | 2.02 | **82.78** |
| ISED (ours) | 80.34 | **95.10** | **94.10** | **97.34** | **82.32** | **79.95** | **68.59** | 80.32 |

of YOLOv8 and only optimize the custom neural network. The neural program solution prompts GPT-4 to classify the scene.

We also consider several tasks from the neurosymbolic literature, including hand-written formula (HWF) evaluation and Sudoku solving. While the solutions to many of these tasks are usually presented as a logic program in neurosymbolic learning frameworks, neural program solutions can take the form of Python programs. We call such models neuroPython programs.

**MNIST-R.** MNIST-R [13, 14] contains 11 tasks operating on inputs of images of handwritten digits from the MNIST dataset [11]. This synthetic test suite includes tasks performing arithmetic ($sum_2$, $sum_3$, $sum_4$, $mult_2$, $mod_2$, add-mod-3, add-sub), comparison (less-than, equal), counting (count-3-or-4), and negation (not-3-or-4) over the digits depicted in the images. Each task dataset has a training set of 5K samples and a testing set of 500 samples.

**HWF.** The goal of the HWF task is to classify images of handwritten digits and arithmetic operators and evaluate the formula [12]. The dataset contains 10K formulas of length 1-7, with 1K length 1 formulas, 1K length 3 formulas, 2K length 5 formulas, and 6K length 7 formulas.

**Visual Sudoku.** The goal of this task is to solve an incomplete 9x9 Sudoku, where the problem board is given as MNIST digits. We follow the experimental setting of NASR [5], including their pre-trained MNIST digit recognition models and sudoku solvers. We use the SatNet dataset consisting of 9K training samples and 500 test samples [25].

## 4.2 Evaluation Setup and Baselines

All of our experiments were conducted on a machine with two 20-core Intel Xeon CPUs, one NVIDIA RTX 2080 Ti GPU, and 755 GB RAM. Unless otherwise noted, the sample count, i.e., the number of calls to the program $P$ per training example, is fixed at 100 for all relevant methods. For additional details on experimental setup, see Appendix B. We apply a timeout of 10 seconds per testing sample, and report the average accuracy and 1-sigma standard deviation obtained from 10 randomized runs.

We pick as baselines neurosymbolic methods DeepProbLog (DPL) [14] and Scallop [13], A-NeSI [24] which performs neural approximation of the gradients, and sampling-based gradient approximation methods REINFORCE [26], IndeCateR [21], and NASR [5]. IndeCateR achieves provably lower variance than REINFORCE by using a specialized sampling method (Appendix A), and NASR is a variant specialized for efficient finetuning by using a single sample and a custom reward function. We also use purely neural baselines and CLIP [19] for GPT leaf and scene. CLIP is a multimodal model that supports zero-shot image classification by simply providing names of the output categories.

## 4.3 RQ1: Performance and Accuracy

To answer **RQ1**, we evaluate ISED's accuracy against those of the baselines. ISED matches, and in many cases surpasses, the accuracy of neurosymbolic and gradient estimation baselines. We highlight the results for $sum_n$ from MNIST-R and other benchmarks in Table 3. Tables 7-10 in Appendix C

Table 4: Performance comparisons for $\text{sum}_8$, $\text{sum}_{12}$, and $\text{sum}_{16}$ with different sample counts $k$.

| | Accuracy (%) | | | | | |
| | $\text{sum}_8$ | | $\text{sum}_{12}$ | | $\text{sum}_{16}$ | |
| Method | $k = 80$ | $k = 800$ | $k = 120$ | $k = 1200$ | $k = 160$ | $k = 1600$ |
|---|---|---|---|---|---|---|
| REINFORCE | 8.32 | 8.28 | 7.52 | 8.20 | 5.12 | 6.28 |
| IndeCateR | 5.36 | **89.60** | 4.60 | 77.88 | 1.24 | 5.16 |
| IndeCateR+ | 10.20 | 88.60 | 6.84 | **86.92** | 4.24 | **83.52** |
| ISED (Ours) | **87.28** | 87.72 | **85.72** | 86.72 | **6.48** | 8.13 |

contain results for the remaining MNIST-R tasks, including standard deviations for all tasks. ISED is the top performer on 8 out of the 16 total tasks.

On the GPT leaf and scene tasks, ISED outperforms the purely neural baseline by $3.82\%$ and $31.42\%$ respectively, and zero-shot CLIP by $59.80\%$ and $17.50\%$. For many tasks, A-NeSI is the non-neurosymbolic method that comes closest to ISED, sometimes outperforming our method. However, A-NeSI achieves significantly lower performance than ISED on tasks involving complex programs, namely HWF and sudoku. This is likely due to the difficulty of training a neural model to estimate the output of $P$ and its gradient when $P$ is complex. ISED also outperforms REINFORCE on all but 3 tasks due to the REINFORCE learning signal being weaker for tasks where $P$ involves multiple inputs. NASR outperforms ISED only on sudoku by $2.46\%$ due to NASR being well-suited for fine-tuning as it restricts its algorithm to use a single sample. IndeCateR achieves similar performance compared to ISED on most tasks but achieves significantly lower accuracy on the scene classification task, which has a large input space with maximum 10 objects each with 47 possible values in each scene, demonstrating that IndeCateR is less sample-efficient than ISED. We elaborate more on this point in **RQ2**.

ISED outperforms the neurosymbolic methods on 9 out of 14 tasks that can be written in logic programming languages. Despite treating $P$ as a black-box, ISED even outperforms Scallop on HWF by $0.69\%$ and comes within $1.16\%$ of NGS, a specialized neurosymbolic learning framework that uses abductive reasoning [12]. Furthermore, DPL timed out on 4 tasks, and Scallop timed out on 1 (sudoku). These results demonstrate that even for tasks that can be written in a logic programming language, treating the program as a black-box can often yield optimal results.

### 4.4 RQ2: Sample Efficiency

To answer **RQ2**, we evaluate the sample efficiency of ISED against REINFORCE, IndeCateR, and IndeCateR+ on adding MNIST digits. IndeCateR+ [21] is a variant of IndeCateR with a sampling method and loss computation customized for higher dimensional setting such as the addition of 16 MNIST digits. We vary the size of the input and output space ($\text{sum}_8$, $\text{sum}_{12}$, $\text{sum}_{16}$) of $P$ as well as the sample count, and report the average accuracy and standard deviation obtained from 5 randomized runs (Tables 4, 11-13).

For a lower number of samples, ISED outperforms all other methods on the three tasks, outperforming IndeCateR by over $80\%$ on $\text{sum}_8$ and $\text{sum}_{12}$. The experimental findings support the conceptual difference of REINFORCE-based methods providing a weak learning signal compared to ISED (Section 3.1). While ISED achieves accuracy similar to the top performer for $\text{sum}_8$ and $\text{sum}_{12}$ with a high sample count, it comes second on $\text{sum}_{16}$ with IndeCateR+ beating ISED by $75.39\%$. This suggests our approach is limited in scaling to high-dimensional inputs to $P$, and motivates exploring better sampling techniques, which is the core difference between IndeCateR and IndeCateR+.

### 4.5 RQ3: Data Efficiency

We now examine how ISED compares to state-of-the-art baselines in terms of data efficiency. We compare ISED and A-NeSI in terms of training time and accuracy on $\text{sum}_3$ and $\text{sum}_4$. We choose these tasks for evaluation because A-NeSI has been shown to scale well to multi-digit addition tasks [24]. Furthermore, these tasks come from the MNIST-R suite in which we use 5K training samples, which is less than what A-NeSI would have used in its evaluation (20K training samples for $\text{sum}_3$

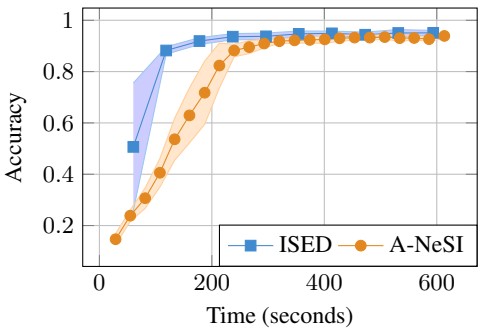

Figure 3: Accuracy vs. Time for $\text{sum}_3$.

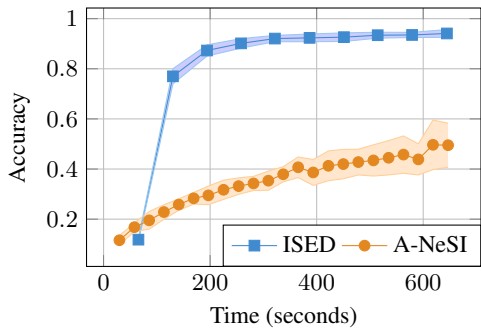

Figure 4: Accuracy vs. Time for $\text{sum}_4$.

and 15K for $\text{sum}_4$). We plot the average test accuracy and standard deviation vs. training time (over 10 runs) in Figures 3 and 4, where each point represents the result of 1 epoch.

While ISED and A-NeSI learn at about the same rate for $\text{sum}_3$ after about 5 minutes of training, ISED learns at a much faster rate for the first 5 minutes, reaching an accuracy of $88.22\%$ after just 2 epochs (Fig. 3). The difference between ISED and A-NeSI is more pronounced for $\text{sum}_4$, with ISED reaching an accuracy of $94.10\%$ after just 10 epochs while A-NeSI reaches $49.51\%$ accuracy at the end of its 23rd epoch (Fig. 4). These results demonstrate that with limited training data, ISED is able to learn more quickly than A-NeSI, even for simple tasks. This result is likely due to A-NeSI training 2 additional neural models in its learning pipeline compared to ISED, with A-NeSI training a prior as well as a model to estimate the program output and gradient.

## 5 Limitations and Future Work

The main limitation of ISED is the difficulty of scaling with the dimensionality of the space of inputs to the program $P$. There are interesting future directions in adapting and expanding ISED for high dimensionality. Specifically, improvements to the sampling strategy could help adapt ISED to a complex space of inputs. Techniques can be borrowed from the field of Bayesian optimization where such large spaces have traditionally been studied. Furthermore, there is merit to systematically combining white-box and black-box methods. ISED is especially useful when logic programs fail to encode reasoning components. Therefore, we believe that ISED can be used as an underlying engine for a new neurosymbolic language that blends the accessibility of black-box with the performance of white-box methods.

## 6 Related Work

**Neurosymbolic programming frameworks.** These frameworks provide a general mechanism to define white-box neurosymbolic programs. DeepProbLog [14] and Scallop [13] abstract away gradient calculations behind a rule-based language. Others specialize in targeted applications, such as NeurASP [28] for answer set programming, or NeuralLog [3] for phrase alignment in NLP. ISED is similar in that it seeks to make classes of neurosymbolic programs easier to write and access; however, it diverges by offering an interface not bound by any specific domain or language syntax.

**RL and sampling-based neurosymbolic frameworks.** ISED incorporates concepts found in the RL algorithm REINFORCE [26] such as the sampling of actions according to the current policy distribution, similar to NASR [5], and IndeCateR [21]. Other work has proposed a semantic loss function for neurosymbolic learning which measures how well neural network outputs match a given constraint [27]. While this technique resembles ISED in that it samples symbols from their predicted distributions to derive the loss, it relies on symbolic knowledge in the form of a constraint in Boolean logic, whereas ISED allows the program component to be any black-box program.

**Specialized neurosymbolic methods.** The majority of the neurosymbolic learning literature pertains to point solutions for specific use cases [7, 25]. In the HWF example, NGS [12] and several of its variants leverage a hand-defined syntax defining the inherent structure within mathematical expressions. Similarly, DiffSort [18] leverages the symbolic properties of sorting to produce differentiable

sorting networks. Other point solutions address broader problem setups, such as NS-CL [15] which provides a framework for visual question answering by learning symbolic representations in text and images. For reading comprehension, the NeRd [3] framework converts NL questions into executable programs over symbolic information extracted from text. ISED aligns with all of these point solutions by aiming to solve problems that have thus far required technically specific solutions in order to access the advantages of neurosymbolic learning, but it takes an opposite and easier approach by forgoing significant specializations and instead leverages existing solutions as black-boxes.

**Differentiable programming and non-differentiable optimization.** Longstanding libraries in deep learning have grown to great popularity for their ability to abstract away automatic differentiation behind easy-to-use interfaces. PyTorch [17] is able to do so by keeping track of a dynamic computational graph. Similarly, JAX [1] leverages functional programming to abstract automatic differentiation. ISED follows the style of these frameworks by offering an interface to abstract away gradient calculations for algorithms used in deep learning, but ISED improves upon them by allowing systematic compatibility of non-differentiable functions.

# 7 Conclusion

We proposed ISED, a data- and sample-efficient algorithm for learning neural programs. Unlike existing general neurosymbolic frameworks which require differentiable logic programs, ISED is compatible with Python programs and API calls to GPT, and it employs a sampling-based technique to learn neural model parameters using forward evaluation. We showed that for neuroGPT, neuroPython, and neurosymbolic benchmarks, ISED achieves better accuracy than end-to-end neural models and similar accuracy compared to neurosymbolic frameworks. ISED also often achieves superior accuracy on complex programs compared to REINFORCE-based and gradient estimation baselines. Furthermore, ISED learns in a more data- and sample-efficient manner compared to these baselines.

# 8 Acknowledgements

We thank the anonymous reviewers for useful feedback. This research was supported by ARPA-H grant D24AC00253-00, NSF award CCF 2313010, and by a gift from AWS AI to ASSET (Penn Engineering Center on Trustworthy AI).

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

# A  Explanation of Differences Between ISED and Baseline Methods

We explain the differences between ISED and prior techniques using the simple example of $\text{sum}_2$, where digits are restricted to be between 0-2. Suppose that we are training a neural network $M_\theta$ for this task, and we are considering the symbol-output sample where the ground truth symbols are 1 and 2, i.e., $r_1 = 1, r_2 = 2$, and the ground truth output is $y = 3$. Suppose that the predicted distributions from $M_\theta$ for $r_1$ and $r_2$ are $[0.1, 0.6, 0.3]$ and $[0.2, 0.1, 0.7]$ respectively. We now explain how different methods perform their loss computations.

## A.1  ISED

Suppose ISED is initialized with a sample count of 3, and the sampled symbol-output pairs are $((1, 2), 3)$, $((1, 0), 1)$, and $((2, 1), 3)$. We use the `add-mult` semiring in this example. ISED can be thought of as differentiating through the following summary logic program:

$$r_1 = 1 \land r_2 = 2 \rightarrow y = 3$$
$$r_1 = 1 \land r_2 = 0 \rightarrow y = 1$$
$$r_1 = 2 \land r_2 = 1 \rightarrow y = 3$$

As a result, the final vector calculated for the loss function before normalization would be

$$\begin{bmatrix} 0.0 \\ 0.6 * 0.2 \\ 0.0 \\ 0.6 * 0.7 + 0.3 * 0.1 \\ 0.0 \end{bmatrix}$$

where each value corresponds to the probability of the given output (possible outputs are in the range 0-4). Note that if there are duplicate samples, ISED includes the duplicate probabilities in its aggregation. In our implementation, we would perform normalization on this vector and then pass it into the binary cross-entropy loss function, with the ground truth vector being:

$$\begin{bmatrix} 0.0 \\ 0.0 \\ 0.0 \\ 1.0 \\ 0.0 \end{bmatrix}$$

We would then minimize this loss and update $M_\theta$ accordingly.

If we use the `min-max` semiring instead, $*$ is replaced by `min` and $+$ by `max` in the final vector calculation, resulting in

$$\begin{bmatrix} 0.0 \\ 0.2 \\ 0.0 \\ 0.6 \\ 0.0 \end{bmatrix}$$

## A.2  REINFORCE

Suppose REINFORCE is also initialized with a sample count of 3, and it samples the same symbol-output pairs. The final reward is computed by element-wise multiplication of the log probability of each sample with its reward value and taking the mean, as follows:

$$\frac{1}{3} * \begin{bmatrix} \log(0.6) + \log(0.7) \\ \log(0.6) + \log(0.2) \\ \log(0.3) + \log(0.1) \end{bmatrix} * \begin{bmatrix} 1.0 \\ 0.0 \\ 1.0 \end{bmatrix}$$

and the goal is to optimize $M_\theta$ to maximize this reward. While this approach resembles ISED's loss computation for the `add-mult` semiring, it does not involve the `mult` step. As it rewards possible values instead of possible combinations, the final reward would have been the same when (1,1) and (2,2) were the correct samples, instead of (1,2) and (2,1). Hence, the learning signal is weaker compared to ISED when there is more than one input to $P$.

### A.3 IndeCateR and NASR

IndeCateR is an extension of the REINFORCE estimator that is unbiased with a provably lower variance. It assumes and exploits the factoring of the underlying multivariate distribution into independent categorical variables by summing out one dimension while keeping a sample for other dimensions fixed. For each sample drawn, IndeCateR systematically creates additional samples that differ on a single entry by enumerating all possible values for each variable. NASR targets efficient finetuning by setting the sample count to one and customizing the reward function.

The loss computation for IndeCateR and NASR are identical to that of REINFORCE, also providing weak signals with fewer samples. Furthermore, both set the reward to 0 for samples leading to incorrect predictions, effectively ignoring them, unlike ISED which penalizes such symbols. Since only the correct symbol contribute to the final reward, the signal is sparser than ISED, making it sample-inefficient.

### A.4 A-NeSI

A-NeSI trains two additional neural networks: a prediction model $Q_{\theta'}$ as a surrogate for $P$, and a prior model $R_\alpha$ for learning the parameters $\alpha$ for the Dirichlet distribution $D_\alpha$.

Suppose A-NeSI is also initialized with a sample count of 3. At each training step, A-NeSI first updates $\alpha$ using $\hat{y} = M_\theta(x)$. Next, it samples a single symbol from each of the 3 distributions sampled from $D_\alpha$, and uses the sampled symbol-output pair and the standard cross entropy loss to update $Q_{\theta'}$. Then, A-NeSI optimizes $M_\theta$ by minimizing the loss $\mathcal{L}(Q_{\theta'}(M_\theta(r_1, r_2)))$ using the prediction model instead of $P$.

### A.5 DeepProbLog

DeepProbLog (DPL) enumerates all possible proofs for each output and aggregates probabilities accordingly. For example, the proofs for $y = 1$ include $r_1 = 0, r_2 = 1$ and $r_1 = 1, r_2 = 0$. Thus, the probability of this output is $0.6 * 0.2 + 0.1 * 0.1$. The final vector calculated would be

$$\begin{bmatrix} 0.1 * 0.2 \\ 0.6 * 0.2 + 0.1 * 0.1 \\ 0.1 * 0.7 + 0.6 * 0.1 + 0.3 * 0.2 \\ 0.6 * 0.7 + 0.3 * 0.1 \\ 0.3 * 0.7 \end{bmatrix}$$

and we would pass this vector into some loss function (e.g., cross-entropy), with the same ground truth vector that ISED would use. DPL would then minimize this loss and update $M_\theta$ accordingly.

### A.6 Scallop

Suppose Scallop is configured to use the `diff-top-1-proofs` semiring. This means that for each possible output, Scallop will use the proof of that output with the highest probability. For instance, the most likely proof for $y = 1$ is $r_1 = 1$ and $r_2 = 0$, and the probability of the output $y = 1$ is $0.6 * 0.2$. The final vector calculated would be

$$\begin{bmatrix} 0.1 * 0.2 \\ 0.6 * 0.2 \\ 0.1 * 0.7 \\ 0.6 * 0.7 \\ 0.3 * 0.7 \end{bmatrix}$$

and we would pass this vector into some loss function (e.g., binary-cross-entropy), with the same ground truth vector that ISED would use. Scallop would then minimize this loss and update $M_\theta$ accordingly. This probability estimation would change depending on the choice of semiring (e.g., `diff-top-k-proofs` for a different value of $k$).

## B   Evaluation Setup

For tasks with the MNIST dataset as unstructured data, we employ LeNet [11], a 2-layer CNN-based model, except for $\text{sum}_8$, $\text{sum}_{12}$, and $\text{sum}_{16}$ tasks where we choose a smaller 2-layer CNN used by

IndeCateR [21]. For HWF, we also use a 2-layer CNN-based model. For leaf classification tasks, images are scaled down and passed to a simple CNN-based network with 4 convolutional layers. For the scene recognition, we use YOLOv8 and a 3-layer convolutional network for neural program methods, 7-layer CNN for the purely neural solution, and CLIP with ViT-B/32. For all tasks included in **RQ1**, other than sudoku, we remove the final softmax function at the end of each network when evaluating IndeCateR since its sampling procedure yields optimal results without the softmax. We also do that same with REINFORCE if it results in higher accuracy. Since sudoku uses a pretrained CNN, we use the same CNN across all methods, including IndeCateR and REINFORCE.

We use the Adam optimizer with the best learning rate among $\{1e-3, 5e-4, 1e-4\}$. We train for maximum 100 epochs, but stop early if the training saturates. For MNIST-R tasks, we used learning rate $1e-4$ and trained ISED for 10 epochs, REINFORCE and IndeCateR for 50 epochs, and A-NeSI and NASR for 100 epochs. We trained ISED for 30 epochs, A-NeSI for 100 epochs, and the rest for 50 epochs for HWF and Leaf Classification with learning rate $1e-4$. For the Scene Recognition task, we trained A-NeSI and the purely neural baseline 50 epochs and the rest 100 epochs with learning rate $5e-4$. For tasks $\text{sum}_8$ to $\text{sum}_{16}$ we trained ISED for 50 epochs and the rest for 100 epochs with learning rate $1e-3$. For Visual Sudoku, we follow the setting in NASR [5] and train for 10 epochs with learning rate $1e-5$.

We configure ISED to use the `min-max` semiring for HWF and the `add-mult` semiring for all other tasks. We use categorical sampling and binary cross-entropy loss for ISED.

### B.1 Neural-GPT Experiment Prompts

For leaf classification and scene recognition, the neural-GPT experiments, we used the up-to-date version of GPT-4, `gpt-4-1106-preview` and `gpt-4o` respectively, with the parameter `top-p` set to $1e-8$. We present the prompts used for the experiments in Tables 5 and 6.

Table 5: GPT-4 prompt for the leaf classification task.

| | |
|---|---|
| **System message** | You are an expert in classifying plant species based on the margin, shape, and texture of the leaves. You are designed to output a single JSON. |
| **User message** | <PLANT NAME>. Classify into one of: <MARGIN/SHAPE/TEXTURE>. Give your answer without explanation. |

Table 6: GPT-4 prompt for the scene recognition task.

| | |
|---|---|
| **System message** | You are an expert at identifying room types based on the object detected. Give short single responses. |
| **User message** | There are <DETECTED OBJECTS>. What type of room is most likely? Choose among <SCENES>. |

We use MARGIN = {entire, dentate, lobed, serrate, serrulate, undulate}, SHAPE ={elliptical, lanceolate, obovate, oblong, ovate}, TEXTURE ={glossy, leathery, smooth, medium}, and PLANT NAME $\in$ {Alstonia Scholaris, Citrus limon, Jatropha curcas, Mangifera indica, Ocimum basilicum, Platanus orientalis, Pongamia Pinnata, Psidium guajava, Punica granatum, Syzygium cumini, Terminalia Arjuna}.

Furthermore, SCENES = {bathroom, bedroom, dining room, living room, kitchen, lab, office, home lobby, basement} and DETECTED OBJECTS is a list of maximum length 10 with duplicates.

## C Full Performance Summary

We report the accuracy of all benchmarks with 1-sigma standard deviation in Tables 7, 8, 9, and 10. We further provide the performance comparison with varying sample counts with 1-sigma standard deviation in Tables 11, 12, and 13.

Table 7: Performance comparison for DT leaf, GPT leaf, scene, and sudoku.

| | Accuracy (%) | | | |
|---|---|---|---|---|
| **Method** | DT leaf | GPT leaf | scene | sudoku |
| DPL | $39.70 \pm 6.55$ | N/A | N/A | TO |
| Scallop | $81.13 \pm 3.50$ | N/A | N/A | TO |
| A-NeSI | $78.82 \pm 4.42$ | $72.40 \pm 12.24$ | $61.46 \pm 14.18$ | $26.36 \pm 12.68$ |
| REINFORCE | $40.24 \pm 0.08$ | $53.84 \pm 0.04$ | $12.17 \pm 0.02$ | $79.08 \pm 0.87$ |
| IndeCateR | $78.71 \pm 5.59$ | $69.16 \pm 2.35$ | $12.72 \pm 2.51$ | $66.50 \pm 1.37$ |
| NASR | $16.41 \pm 1.79$ | $17.32 \pm 1.92$ | $2.02 \pm 0.23$ | $\mathbf{82.78 \pm 1.06}$ |
| ISED (ours) | $\mathbf{82.32 \pm 4.15}$ | $\mathbf{79.95 \pm 5.71}$ | $\mathbf{68.59 \pm 1.95}$ | $80.32 \pm 1.79$ |

Table 8: Performance comparison for HWF, $\mathrm{sum}_2$, $\mathrm{sum}_3$, and $\mathrm{sum}_4$.

| | Accuracy (%) | | | |
|---|---|---|---|---|
| **Method** | HWF | $\mathrm{sum}_2$ | $\mathrm{sum}_3$ | $\mathrm{sum}_4$ |
| DPL | TO | $95.14 \pm 0.80$ | $93.80 \pm 0.54$ | TO |
| Scallop | $96.65 \pm 0.13$ | $91.18 \pm 13.43$ | $91.86 \pm 1.60$ | $80.10 \pm 20.4$ |
| A-NeSI | $3.13 \pm 0.72$ | $\mathbf{96.66 \pm 0.87}$ | $94.39 \pm 0.77$ | $78.10 \pm 19.0$ |
| REINFORCE | $88.27 \pm 0.02$ | $74.46 \pm 26.29$ | $19.40 \pm 4.52$ | $13.84 \pm 2.26$ |
| IndeCateR | $95.08 \pm 0.41$ | $96.48 \pm 0.53$ | $93.76 \pm 0.47$ | $92.58 \pm 0.80$ |
| NASR | $1.85 \pm 0.27$ | $6.08 \pm 0.77$ | $5.48 \pm 0.77$ | $4.86 \pm 0.93$ |
| ISED (ours) | $\mathbf{97.34 \pm 0.26}$ | $80.34 \pm 16.14$ | $\mathbf{95.10 \pm 0.95}$ | $\mathbf{94.1 \pm 1.6}$ |

Table 9: Performance comparison for $\mathrm{mult}_2$, $\mathrm{mod}_2$, less-than, and add-mod-3.

| | Accuracy (%) | | | |
|---|---|---|---|---|
| **Method** | $\mathrm{mult}_2$ | $\mathrm{mod}_2$ | less-than | add-mod-3 |
| DPL | $95.43 \pm 0.97$ | $96.34 \pm 1.06$ | $\mathbf{96.60 \pm 1.02}$ | $95.28 \pm 0.93$ |
| Scallop | $87.26 \pm 24.70$ | $77.98 \pm 37.68$ | $80.02 \pm 3.37$ | $75.12 \pm 21.64$ |
| A-NeSI | $96.25 \pm 0.76$ | $96.89 \pm 0.84$ | $94.75 \pm 0.98$ | $77.44 \pm 24.60$ |
| REINFORCE | $\mathbf{96.62 \pm 0.23}$ | $94.40 \pm 2.81$ | $78.92 \pm 2.31$ | $\mathbf{95.42 \pm 0.37}$ |
| IndeCateR | $96.32 \pm 0.50$ | $\mathbf{97.04 \pm 0.39}$ | $94.98 \pm 1.50$ | $78.52 \pm 23.26$ |
| NASR | $5.34 \pm 0.68$ | $20.02 \pm 2.67$ | $49.30 \pm 2.14$ | $33.38 \pm 2.81$ |
| ISED (ours) | $96.02 \pm 1.13$ | $96.68 \pm 0.93$ | $96.22 \pm 0.95$ | $83.76 \pm 12.89$ |

Table 10: Performance comparison for add-sub, equal, not-3-or-4, and count-3-4.

| | Accuracy (%) | | | |
|---|---|---|---|---|
| **Method** | add-sub | equal | not-3-or-4 | count-3-4 |
| DPL | $93.86 \pm 0.87$ | $\mathbf{98.53 \pm 0.37}$ | $98.19 \pm 0.55$ | TO |
| Scallop | $92.02 \pm 1.58$ | $71.60 \pm 2.29$ | $97.42 \pm 0.73$ | $93.47 \pm 0.83$ |
| A-NeSI | $93.95 \pm 0.60$ | $77.89 \pm 36.01$ | $98.63 \pm 0.50$ | $93.73 \pm 2.93$ |
| REINFORCE | $17.86 \pm 3.27$ | $78.26 \pm 3.96$ | $\mathbf{99.28 \pm 0.21}$ | $87.78 \pm 1.14$ |
| IndeCateR | $93.74 \pm 0.44$ | $98.18 \pm 0.39$ | $99.26 \pm 0.16$ | $94.30 \pm 1.26$ |
| NASR | $5.26 \pm 1.10$ | $81.72 \pm 1.94$ | $68.36 \pm 1.54$ | $25.26 \pm 1.66$ |
| ISED (ours) | $\mathbf{95.32 \pm 0.81}$ | $96.02 \pm 1.74$ | $98.08 \pm 0.72$ | $\mathbf{95.26 \pm 1.04}$ |

Table 11: Performance comparison for $\text{sum}_8$ with different sample counts $k$.

| | Accuracy (%) | |
| --- | --- | --- |
| | $\text{sum}_8$ | |
| **Method** | $k = 80$ | $k = 800$ |
| REINFORCE | $8.32 \pm 2.52$ | $8.28 \pm 0.39$ |
| IndeCateR | $5.36 \pm 0.26$ | $\mathbf{89.60 \pm 0.98}$ |
| IndeCateR+ | $10.20 \pm 1.12$ | $88.60 \pm 1.09$ |
| ISED (Ours) | $\mathbf{87.28 \pm 0.76}$ | $87.72 \pm 0.86$ |

Table 12: Performance comparison for $\text{sum}_{12}$ with different sample counts $k$.

| | Accuracy (%) | |
| --- | --- | --- |
| | $\text{sum}_{12}$ | |
| **Method** | $k = 120$ | $k = 1200$ |
| REINFORCE | $7.52 \pm 1.92$ | $8.20 \pm 1.80$ |
| IndeCateR | $4.60 \pm 0.24$ | $77.88 \pm 6.68$ |
| IndeCateR+ | $6.84 \pm 2.06$ | $\mathbf{86.92 \pm 1.36}$ |
| ISED (Ours) | $\mathbf{85.72 \pm 2.15}$ | $86.72 \pm 0.48$ |

Table 13: Performance comparison for $\text{sum}_{16}$ with different sample counts $k$.

| | Accuracy (%) | |
| --- | --- | --- |
| | $\text{sum}_{16}$ | |
| **Method** | $k = 160$ | $k = 1600$ |
| REINFORCE | $5.12 \pm 1.91$ | $6.28 \pm 1.04$ |
| IndeCateR | $1.24 \pm 1.68$ | $5.16 \pm 0.52$ |
| IndeCateR+ | $4.24 \pm 0.95$ | $\mathbf{83.52 \pm 1.75}$ |
| ISED (Ours) | $\mathbf{6.48 \pm 0.50}$ | $8.13 \pm 1.10$ |

## D    License Information

For implementing the baselines, we adapted the code from the official repositories of DeepProblog [14] (Apache 2.0), Scallop [8] (MIT), A-NeSI [24] (MIT), NASR [5] (MIT), and IndeCateR [21] (Apache 2.0). Additionally, our benchmarks use Multi-illumination dataset [16] (CC BY 4.0), HWF dataset (CC BY-NC-SA 3.0) from NGS [12], a subset of the leaf database [4] (CC BY 4.0), YOLOv8 (AGPL-3.0) and CLIP (MIT).

