# OpenReview forum: "Data-Efficient Learning with Neural Programs"
_NeurIPS.cc/2024/Conference — NeurIPS 2024 poster_

### Official Review · Reviewer_5P1v · 2024-06-29

**Soundness:** 3
**Presentation:** 2
**Contribution:** 3
**Rating:** 6
**Confidence:** 2

**Summary:**

This paper looks at optimizing "neural programs", which are composites of a deep neural network (DNN) followed by a program written in a traditional programming language or an API call to a large language model (LLM). This paper proposes ISED (Infer-Sample-Estimate-Descend) for learning neural programs that only rely on input-output samples of black-box components. ISED is evaluated on benchmarks involving calls to GPT-4 and prior work from neurosymbolic learning literature, showing comparable performance to state-of-the-art neurosymbolic frameworks and achieving more data- and sample-efficient results.

**Strengths:**

* From comprehensive evaluations, ISED significantly outperforms prior work regarding accuracy and data efficiency.
* The proposed ISED approach is novel by estimating gradients of black-box programs from inductive samples.

**Weaknesses:**

* Clarity: While Section 1 and 2 is nicely written and can be understood by readers new to this field, it is challenging to read Section 3 due to over-formalism and the lack of end-to-end running examples.
* Clarity: The symbolism is a bit inconsistent. $x$ and $y$ in Section 2 presents the input to $M_\theta$ and output from $P$, while in Section 3.1 $x$ and $y$ denotes the "input-output samples" of $P$.

Other minor editorial suggestions:
* L3 and L20: double quotation can be done via `` and '' in LaTeX.

**Questions:**

* What are the choice reasons for the types of defined structural mapping?
* What does "interpretation" mean here in L145? How is that related?
* Besides scalability, what other future directions do the authors see as promising for the development and improvement of neural programs?

**Limitations:**

As is acknowledged by the authors, ISED struggles with scaling to high-dimensional input spaces, limiting its applicability in complex scenarios.

---

> ### Author Rebuttal · Authors · 2024-08-06
>
> > While Section 1 and 2 is nicely written and can be understood by readers new to this field, it is challenging to read Section 3 due to over-formalism and the lack of end-to-end running examples.
>
> Thank you for this feedback. We agree that the lack of end-to-end running examples makes the algorithm harder to understand and will keep using the HWF example throughout our explanation in the final version.
>
> > The symbolism is a bit inconsistent. and in Section 2 presents the input to and output from , while in Section 3.1 and denotes the "input-output samples" of .
>
> Thank you for this suggestion. We noticed this inconsistency after submission and improved the wording in a newer draft of the paper.
>
> > What are the choice reasons for the types of defined structural mapping?
>
> We define these structural mappings to state what input and output types of black-box programs are supported by our framework. Since Python programs support many kinds of structured inputs, we wanted to formulate ISED in a way that is compatible with a diverse set of programs. For example, ISED supports sorting variable length lists of digits (hence the permutation type), which is a task that cannot be easily expressed in other neurosymbolic learning frameworks like Scallop [1]. The types defined in the paper is not an exhaustive list of types supported by the framework, but rather a fraction that were implemented for our benchmarks. The structural mappings also serve to exchange information between the neural network and the black-box program, and explain how to derive the loss, which is why we define vectorizer and aggregator for each type.
>
> [1] Scallop: From Probabilistic Deductive Databases to Scalable Differentiable Reasoning, NeurIPS 2021
>
> > What does "interpretation" mean here in L145? How is that related?
>
> “Interpretation” refers to the discrete vs probabilistic interpretations of inputs and outputs to the black-box program. We distinguish the two interpretations to make it clear how the sampling stage deals with probability distribution (from the neural network) and samples discrete values from them. Additionally, we wanted to clearly explain how the discrete outputs from the black-box program are vectorized/ turned into probabilistic outputs, which is crucial for computing the loss.
>
> > Besides scalability, what other future directions do the authors see as promising for the development and improvement of neural programs?
>
> For other future work, we would like to extend the gradient estimation technique to be compatible with more complex architectures. For example, end-to-end training with a neural network, followed by a program, followed by another neural network could be an interesting direction, especially if the final neural network is trying to train a bias correction parameter. Such an architecture would require an explicit estimation of the Jacobian of the black-box program, which ISED does not currently do. This estimation could use finite difference or randomized finite difference techniques from numerical analysis. For this work, we plan to also provide some theoretical justification behind the gradient estimation, possibly showing that it is an unbiased estimator of the true gradient.
>
> > ISED seems challenging to implement and understand by practitioners without a strong background in related fields.
>
> In most cases, a practitioner would only need to specify the input and output types of the program component. Furthermore, a benefit of using ISED is that the program can be specified in Python, which removes the need to learn specialized languages like Datalog. However, it is true that it might be challenging to implement a task where the black-box component has an input or output type that our implementation does not yet support.

---

> > ### Comment · Area_Chair_293c · 2024-08-07
> >
> > Dear reviewer 5P1v, could you comment to which degree the author response addressed your concerns? I would be particularly interested in your review on the limitation "ISED seems challenging to implement and understand by practitioners without a strong background in related fields." Thank you!

---

> > ### Comment · Reviewer_5P1v · 2024-08-08
> >
> > Thanks for the detailed explanation. I am now less concerned about the complexity of the user interface in ISED. Therefore, I have increased my rating. Looking forward to the writing improvement in the next revision. Thanks.

---

> > > ### Comment · Area_Chair_293c · 2024-08-08
> > >
> > > Thank you very much for your response!

---

### Official Review · Reviewer_E3jF · 2024-07-12

**Soundness:** 3
**Presentation:** 3
**Contribution:** 3
**Rating:** 5
**Confidence:** 3

**Summary:**

This paper proposes an algorithm ISED for learning a composition between a neural model (e.g., using DNN to classify objects in an image) and traditional programming (e.g., calling GPT-4 to identify room type). ISED is based on reinforcement learning, which summarizes the input-output samples of the traditional black-box program as if-then logic programming. Evaluation of several benchmark tasks, such as leaf classification and scene recognition, shows the effectiveness and generalizability of the ISED.

**Strengths:**

+ Proposing a data-efficient learning framework ISED.
+ Evaluating the effectiveness of ISED on three types of benchmark tasks.
+ Analyzing the efficiency of data sampling and used data for ISED.

**Weaknesses:**

- The motivation could be made stronger.
- The advantages of ISED over individual programming are not investigated.
- Lack of investigation on the impact of the capability of traditional programming on the ISED’s performance.

More specifically:

. (Introduction) The motivation of this work is not very strong. If the two models work well, why composite them with a neural program? For the scene recognition example, DNN plus GPT-4 can identify objects in the image and classify the scene. What is the goal of the composed program?

. (Method) ISED considers traditional programming as a black-box program. It is assumed that ISED can learn from any program. In the evaluation, GPT-4 is a powerful LLM that can generate good outputs. Thus, for a powerful black-box program, ISED works better. It is suggested to analyze how the capability of a black-box program affects the performance of neural symbolic learning.

. (Evaluation) This study presents two new neural program learning benchmarks which both contain a program component that can make a call to GPT-4. A neural model can only learn  limited capability of LLM. The composite program may not replace individual programming?

.(RQ1: Accuracy) The results indicate that ISED can achieve higher performance than other neurosymbolic methods. It is suggested to analyze the differences between the composite program and the individual programming.

. (RQs) The three investigated RQs only focused on the performance and efficiency of ISED. Generally, the composite neural program aims to build a more explainable logic programming. The explainability is not verified.

. The acronyms need to be defined in the first place, such as ISED and LLM.

**Questions:**

- What is the motivation of the composition?
- What are the advantages of ISED over individual programming ?
- What is the impact of the capability of traditional programming on the ISED’s performance?

**Limitations:**

The authors adequately addressed the limitations of this work.

---

> ### Author Rebuttal · Authors · 2024-08-06
>
> To address these concerns, we ran additional experiments (highlighted in bold) for tasks involving GPT-4 which we summarize here.
>
> ### Leaf Classification
>
> | Architecture      | Test Accuracy      |
> | ------------- | ------------- |
> | Purely Neural | 78.50% |
> | CLIP   | 20.15% |
> |  **GPT-4o Vision**  |  52.72% |
> |  **GPT + GPT** |  32.74% |
> |  ISED (GPT) | 79.95% |
> | ISED (DT) |  82.32% |
>
> - GPT-4o vision and CLIP are pre-trained vision-language models
> - GPT+GPT refers to prompting GPT-4o vision to identify leaf traits (margin/shape/texture) from image and asking GPT-4 to classify based on these features.
>
>
> ### Scene Recognition
>
> | Architecture      | Test Accuracy      |
> | ------------- | ------------- |
> | Purely Neural | 37.17% |
> | CLIP  | 51.09% |
> | **YOLO + GPT** | 18.57% |
> | **GPT + GPT** | 32.72% |
> | ISED | 68.59% |
>
> - YOLO+GPT refers to using YOLO, a pre-trained object detector, to identify objects in the scene and asking GPT-4 to classify based on these objects.
> - GPT+GPT refers to prompting GPT-4o to identify objects in the scene and asking GPT-4 to classify the scene.
>
> We now address your individual concerns below.
>
> > What is the motivation of the composition?
>
> Neural programs target tasks that can be better solved with a pipeline involving a DNN followed by a program, compared to DNN alone. The goal of the learning algorithm is to train the neural network in this composed model with only end-to-end labels. The lack of intermediate labels makes it impossible to train the DNN directly and requires a way to backpropagate the loss through the program component. For the scene recognition example, we know the ground truth room type, but not the list of objects found in the scene. Hence, we are unable to train the DNN to detect objects using the standard supervised learning. On the other hand, with ISED, we can train the DNN to detect objects in this DNN plus GPT-4 pipeline, where GPT-4 identifies the room type based on the list of detected objects. A neural component is necessary for this task since traditional (purely symbolic) programs cannot deal with unstructured inputs like images. While we can train a custom DNN or use a pre-trained vision language model to directly predict the room type, we showed in our paper that ISED outperforms the purely neural baseline by 31.42% and zero-shot CLIP by 17.50% respectively.
>
> *Why can we not use only pre-trained models? Why do we need to train with ISED?*
>
> Alternatively, we can use pre-trained models for object detection without any finetuning while keeping the DNN plus GPT-4 structure. Simply using a pre-trained object detector YOLO, and GPT-4 for object detection, resulted in 19.57% and 59.78% accuracy respectively, compared to 68.59% for ISED. This demonstrates the benefit of training to achieve high accuracy on specific tasks.
>
> > What are the advantages of ISED over individual programming?
>
> For the benchmarks involving GPT-4, we also used purely neural networks, and the pre-trained vision language model CLIP, as baselines. We reported that on the leaf classification task, ISED outperforms the purely neural baseline by 3.82% and zero-shot CLIP by 59.80%. For better comparison with the state-of-the-art foundation models, we ran more experiments using the recent vision language model GPT-4o. When directly asked to predict the species from the leaf image, GPT-4o achieves 52.72% accuracy. We also tried a GPT+GPT structure where we ask GPT-4o to predict leaf features from the input image and again prompt GPT-4 to classify based on them, which resulted in 32.73% accuracy. ISED’s GPT Leaf architecture that instead trains a custom neural network to identify leaf features achieves 79.95% accuracy, outperforming both. This demonstrates that it can be useful to compose a neural network and a black-box program to perform reasoning.
>
> > What is the impact of the capability of traditional programming on the ISED’s performance?
>
> Indeed, the capability of a black-box program affects the performance of neurosymbolic learning. This was shown in our two different methods of solving the leaf classification task. One implementation uses a decision tree from a leaf identification guide, and the other prompts the GPT-4 to identify the leaf based on the predicted features (margin, shape, texture). The ISED decision tree and GPT methods achieve 82.32% and 79.95% accuracy, respectively. ISED learns better from the decision tree, which is directly based on expert knowledge and has better leaf classification capability compared to GPT-4. Although it is possible that a newer version of GPT could close this accuracy gap, we would like to note that a good selection of each component of the composite is important for good performance. To benefit from using an LLM in the black-box program, the part delegated to the LLM should be something it can do well. If the LLM produces bad outputs more often than good outputs, ISED wouldn’t be able to learn from such programs.
>
> > Generally, the composite neural program aims to build a more explainable logic programming. The explainability is not verified.
>
> While explainability has not been the focus of our paper, ISED by design offers better explainability than purely neural models. With ISED, we can recover intermediate labels during test time, and there have been prior works [1]  on how intermediate labels improve explainability. Furthermore, when the black-box program is a logic program or a simple Python program without API calls, we can look at the implementation of the symbolic component and analyze how the symbols predicted by the neural network contribute to the final prediction, offering additional explainability.
>
> [1] Concept bottleneck Models, ICML 2020.
>
> > The acronyms need to be defined in the first place, such as ISED and LLM.
>
> In the final version, we will define these acronyms upfront.

---

> > ### Comment · Area_Chair_293c · 2024-08-07
> >
> > Dear authors, thank you very much for the detailed response and the new experiments in particular. Dear reviewer E3jF, could you comment to which degree the response addressed your concerns? Your insight would be appreciated, thank you!

---

> > ### Comment · Reviewer_E3jF · 2024-08-08
> >
> > Thanks for the rebuttal, which partially addressed my concerns.

---

### Official Review · Reviewer_2rwx · 2024-07-13

**Soundness:** 4
**Presentation:** 3
**Contribution:** 3
**Rating:** 7
**Confidence:** 4

**Summary:**

The paper introduces a sample-efficient algorithm called ISED (Iterative Sampling for Efficient Differentiable Programming) for learning neural programs i.e. programs that have a neural network component (that needs to be learned) and another fixed program. The fixed program could be a python program, a call to a black box LLM model, or a logic program. The task here is to train this neural program using only end-to-end input-output data. The challenge lies in propagating gradients through the fixed program, which could be complex, non-differentiable, and black-box. The paper proposes an alternative to REINFORCE-based methods by  trying to approximate the fixed program as if input = u then output = v by sampling multiple u with the NN's predicted distribution and using  neurosymbolic methods to compute gradients through this approximate program.

**Strengths:**

- Combining neural networks with pre-trained LLMs is an interesting idea and very useful in many applications.

- The paper addresses the challenging problem of propagating gradients through black-box LLMs with an interesting approach.

- The proposed method offers an alternative to REINFORCE-based methods which require significantly more samples due to weak supervision.

- The evaluation is comprehensive. The authors considered many variants of benchmarks and baselines. The results are promising, showing better performance and sample efficiency than REINFORCE-based methods and faster training than the A-NeSI baseline.

**Weaknesses:**

One of the main limitations, as mentioned by the authors, is that the approach doesn't work effectively on high-dimensional intermediate inputs. The authors should definitely explore this limitation further.


Also one minor comment: The paper should present results for all benchmarks in Table 3. It is misleading to show mostly positive benchmarks in the main paper and relegate the negative ones to the appendix.

**Questions:**

Why do tests time out during test time? While it is understandable for Sudoku, why does sum_4 also time out for DPL?

**Limitations:**

Limitations are addressed.

---

> ### Author Rebuttal · Authors · 2024-08-06
>
> > The paper should present results for all benchmarks in Table 3. It is misleading to show mostly positive benchmarks in the main paper and relegate the negative ones to the appendix.
>
> We did not intend to be misleading, but we understand how we gave off that impression. We note that this paper goes further than most other neurosymbolic work by evaluating on benchmarks that cannot be expressed as logic programs. Our goal for Table 3 was to highlight these benchmarks as well as the most interesting/complex benchmarks in our suite that were taken from the neurosymbolic literature (sum4, HWF, and Sudoku). In the final version, we will highlight more than just the six benchmarks in the main body to give a better idea of how the techniques compare across the benchmarks.
>
> > Why do tests time out during test time? While it is understandable for Sudoku, why does sum_4 also time out for DPL?
>
> We used DeepProbLog with exact inference, which means that it is still collecting all proofs for each possible output in sum_4. DeepProbLog timing out on this task is consistent with prior work comparing Scallop to DeepProbLog [1] (Table 1).
>
> [1] Scallop: From Probabilistic Deductive Databases to Scalable Differentiable Reasoning, NeurIPS 2021

---

### Official Review · Reviewer_ubXN · 2024-07-13

**Soundness:** 3
**Presentation:** 4
**Contribution:** 3
**Rating:** 7
**Confidence:** 4

**Summary:**

This paper is about learning the parameters for a neural network (or other differentiable structure) where its output is then provided to a different black-box procedure, and we can compute a loss for the (second) output from that procedure.

The procedure consists of four parts: infer, sample (+ execute), and estimate, and descend. Given the input, we "infer" using the neural network, which produces a distribution over the inputs to the black-box program. We "sample" from this distribution multiple times to construct possible inputs to the black-box program, and we run the program on these inputs. We "estimate" by combining the outputs from the program to construct a distribution over the black-box program's output space, only using differentiable operations over the neural network's outputs. We "descend" by computing the loss using that distribution against the ground-truth label and computing the derivative with respect to the neural network's parameters.

The authors consider several benchmark tasks: classifying leaf images to the correct plant species, scene recognition from images, arithmetic and sudoku solving using images of handwritten digits. The authors compare against two neuro-symbolic methods, A-NeSI which emulates the black-box program with a differentiable neural network, and three other methods for computing gradients through black-box programs. The authors show that their proposed method generally outperforms the comparison methods and demonstrates better data efficiency.

**Strengths:**

### Originality
In my view, the paper explores an interesting point in the design space that is an interesting variant some of the existing baselines, e.g. it is akin to a sampled version of DeepProbLog. Much of the value of the paper is in showing that this variant works well empirically on a variety of tasks.

### Quality
The paper performs a thorough evaluation on a variety of different tasks involving different image recognition problems and compares against many baselines, which convincingly demonstrates the empirical gains of this approach.

### Clarity
The paper is generally well-written and easy to understand. Some aspects of the algorithm description could have been clearer, but I found the worked example in Appendix A very useful for both understanding the proposed algorithm and also how it compares to the baselines studied.

**Weaknesses:**

- The studied tasks all have a relatively small space of potential discrete outputs from the black-box program. The method seems to lack support for the case where outputs of the program can be real-valued.
- If the combination of the initial neural network and black-box program produce the correct answer for an input with very low likelihood, or if the sampling is unlucky and fails to produce a black-box program input which gives the correct answer, it would seem that there is no learning signal at all. The method as stated is not adaptive to such cases, for example by sampling from the neural network more often until some suitable input is found.
- The study of the proposed method is entirely empirical and there are no theoretical justifications provided for why it works (also as stated by the authors in item 3 of the paper checklist).

**Questions:**

- What is the specific sampling strategy used in Algorithm 1? Are there restrictions on what this sampling strategy is allowed to be?
- For computing the results in the paper, how did you sample from the neural network at test time?

**Limitations:**

The author's description of the limitations seems adequate.

---

> ### Author Rebuttal · Authors · 2024-08-06
>
> > The method seems to lack support for the case where outputs of the program can be real-valued.
>
> We support program outputs that are real-valued, and the handwritten formula (HWF) benchmark is an example of this. As we mention on L136 and L158, after the sampling step, floating-point output space is discretized by grouping program outputs that are approximately equal (to account for the floating point error) into buckets. We treat these buckets as we would treat the discrete output space in any other task.
>
> > If the combination of the initial neural network and black-box program produces the correct answer for an input with very low likelihood, or if the sampling is unlucky and fails to produce a black-box program input which gives the correct answer, it would seem that there is no learning signal at all.
>
> We first note that ISED works by rewarding sampled symbols that result in the correct output as well as penalizing sampled symbols that result in the wrong output. Even with no positive learning signal, ISED still provides some negative feedback. Although more advanced sampling strategies may reduce the likelihood of failing to produce input that gives the correct answer, any sampling-based strategy will have this limitation. This point is related to the main limitation of ISED that we discuss: it does not scale to high-dimensional inputs because it cannot produce a good learning signal in such cases. Nonetheless, we do intend to address this limitation in future work.
>
> > The study of the proposed method is entirely empirical and there are no theoretical justifications provided for why it works.
>
> We will list this as a limitation.
>
> > What is the specific sampling strategy used in Algorithm 1?
>
> The algorithm is generic and any sampling strategies could be used, but we only used categorical sampling in our experiments.
>
> > For computing the results in the paper, how did you sample from the neural network at test time?
>
> Similar to training time, we perform the steps of ISED and obtain an output distribution. We take the argmax of this distribution as the final prediction.

---

### Author Rebuttal · Authors · 2024-08-06

We thank the reviewers for their insightful comments. Our work explores the idea of composing DNNs with any program that performs reasoning over the network’s predictions. We introduced a novel algorithm for learning such composites with a black-box program component. We introduced new benchmarks that use an LLM in the program component, and our experiments show that our algorithm can learn more effectively than previous techniques in these settings.

We also thank the reviewers for pointing out some weaknesses including the scalability of our algorithm as well as our evaluation that is strictly empirical. These insights help motivate future work on ISED, including improving its scalability and developing a gradient estimator that is provably unbiased.

---

### Decision · Program_Chairs · 2024-09-25

**Decision:**

Accept (poster)

**Comment:**

The paper appears to make significant progress in the direction of neurosymbolic approaches and integration of differential and traditional programming. All reviewers agreed that the novelty in the paper is substantial and that the empiric evaluation is sound. While some technical concerns were raised in the reviews, these could be mostly addressed in the rebuttal phase. Further, the presented type of neuro-symbolic research should be of broad interest to the NeurIPS community. Overall, the clear consensus is to accept the paper.